# Improved sleep, cognitive processing and enhanced learning and memory task accuracy with Yoga nidra practice in novices

Karuna Datta[1]*, Anna Bhutambare[2], Mamatha V. L.[3], Yogita Narawa[2], Rajagopal Srinath[4], Madhuri Kanitkar[5]

1 Human Sleep Research Lab, c/o Department of Sports Medicine, Armed Forces Medical College, Pune, Maharashtra, India, 2 JRF, DST SATYAM Project, Human Sleep Research Lab, c/o Department of Sports Medicine, Armed Forces Medical College, Pune, Maharashtra, India, 3 DST SATYAM Project, Human Sleep Research Lab, c/o Department of Sports Medicine, Armed Forces Medical College, Pune, Maharashtra, India, 4 Department of Internal Medicine, Armed Forces Medical College, Pune, Maharashtra, India, 5 Maharashtra University of Health Sciences, Nashik, Maharashtra, India

* karunadatta@gmail.com

**Data Availability Statement:** All relevant data are within the paper and its Supporting information files.

## Abstract

Complementary and Alternative medicine is known to have health benefits. Yoga nidra practice is an easy-to-do practice and has shown beneficial effects on stress reduction and is found to improve sleep in insomnia patients. Effect of yoga nidra practice on subjective sleep is known but its effect on sleep and cognition objectively is not documented. The aim of the study was to study the effect of yoga nidra practice on cognition and sleep using objective parameters. 41 participants were enrolled, and baseline sleep diary (SD) collected. Participants volunteered for overnight polysomnography (PSG) and cognition testing battery (CTB) comprising of Motor praxis test, emotion recognition task (ERT), digital symbol substitution task, visual object learning task (VOLT), abstract matching (AIM), line orientation task, matrix reasoning task, fractal-2-back test (NBACK), psychomotor vigilance task and balloon analog risk task. Baseline CTB and after one and two weeks of practice was compared. Power spectra density for EEG at central, frontal, and occipital locations during CTB was compared. Repeat SD and PSG after four weeks of practice were done. After yoga nidra practice, improved reaction times for all cognition tasks were seen. Post intervention compared to baseline (95%CI; $p$-value, effect size) showed a significant improvement in sleep efficiency of +3.62% (0.3, 5.15; $p$ = 0.03, $r$ = 0.42), -20min (-35.78, -5.02; $p$ = 0.003, $d$ = 0.84) for wake after sleep onset and +4.19 $\mu V^2$ (0.5, 9.5; $p$ = 0.04, $r$ = 0.43) in delta during deep sleep. Accuracy increased in VOLT (95% CI: 0.08, 0.17; $p$ = 0.002, $d$ = 0.79), AIM (95% CI: 0.03, 0.12; $p$ = 0.02, $d$ = 0.61) and NBACK (95% CI: 0.02, 0.13; $p$ = 0.04, $d$ = 0.56); ERT accuracy increased for happy, fear and anger (95% CI: 0.07, 0.24; $p$ = 0.004, $d$ = 0.75) but reduced for neutral stimuli (95% CI: -0.31, -0.12; $p$ = 0.04, $r$ = 0.33) after yoga nidra practice. Yoga Nidra practice improved cognitive processing and night-time sleep.

**Funding:** KD received the funding from Department of Science and Technology under Science and Technology for Yoga and Meditation (SATYAM), India available at https://dst.gov.in/, and was received vide their sanction order DST Satyam 2018/457 dated 28 May 2020. The funders had no role in study design, data collection and analysis, decision to publish, or preparation of the manuscript.

**Competing interests:** The authors have declared that no competing interests exist.

## 1. Introduction

Complementary and Alternative medicine is known to have benefits in both health and disease [1]. A concept of therapeutic yoga has thus emerged, and a need to build standardised models has become vital. Studies have shown that yoga can be performed in many different ways [2, 3]. However ease of doing yoga is important for its acceptability by the population. Moreover, if any particular yogic practice can be standardised, then it can be easily taught and learnt, thus increasing its effectiveness in promoting wellbeing.

Yoga nidra practice, a kind of *pratyahara* (withdrawal of senses) technique, is an easy-to-do practice [4]. It has been described as an 'awake aware sleep' and thus differs from meditation and hypnosis [4, 5]. Yoga nidra is done in supine posture unlike meditation, which is practiced in seated posture. The practice of yoga nidra in novices showed increased power spectra density of delta frequency band at different areas, like local sleep, during some parts of yoga nidra, while the subject epochs were documenting wake state [6]. This makes yoga nidra unique since it exhibits delta propensity during the practice, unlike meditation which exhibits alpha-theta state effects [4, 7]. Its use has been associated with stress reduction and its role in management of insomnia, menstrual abnormalities, post-traumatic stress disorder, etc and for improved sleep and performance in sportsperson has been documented [8–13]. Yoga nidra practice has been standardised for learning by novices [14], thus improving the effectiveness of its sessions.

Good sleep quality is essential for human performance. Individuals who need to take critical decisions require optimized cognitive functioning. There is evidence that improved sleep enhances cognitive functioning whereas partial sleep deprivation reduces it [15].

Effects of yoga nidra in meditators has been reported earlier [16–19]. However, the effects seen in skilled meditators cannot be compared with those in novices. It is also likely that the effects of meditation itself will confound the effects of yoga nidra in skilled meditators. The methodology for yoga nidra practice by novices has already been laid out [4] and it has been proven to improve sleep in insomnia patients [8, 14]. Novices also reported a subjective improvement of night-time sleep [6, 9, 20] along with an evidence of local sleep during morning yoga nidra practice [6]. Subjective improvement in cognitive scores and self observation notes has also been found using various duration and frequency of yoga nidra practice [20, 21]. However its effect on objective parameters of sleep and cognition in healthy novices is not documented. There was thus a felt need to study the effect of yoga nidra practice by novices on their sleep and cognition objectively.

## 2. Methods

### 2.1 Study design and participants

The project was a pre-post interventional cohort study that aimed to study the effect of yoga nidra practice on cognitive function and sleep.

The study was conducted as a pilot project on healthy volunteers and their cognition was assessed at baseline and after yoga nidra practice. Objective sleep parameters using overnight polysomnography (PSG) were also studied at baseline and after yoga nidra practice.

The yoga nidra supervised training model for novices developed by Datta et al [8, 14] was used for this study. Participants were assessed after one and two weeks of practicing yoga nidra. Details of study design are given in Fig 1(A) and Fig 1(B) for both objectives, i.e., the effect of yoga nidra practice on cognitive function and on night-time objective sleep parameters respectively.

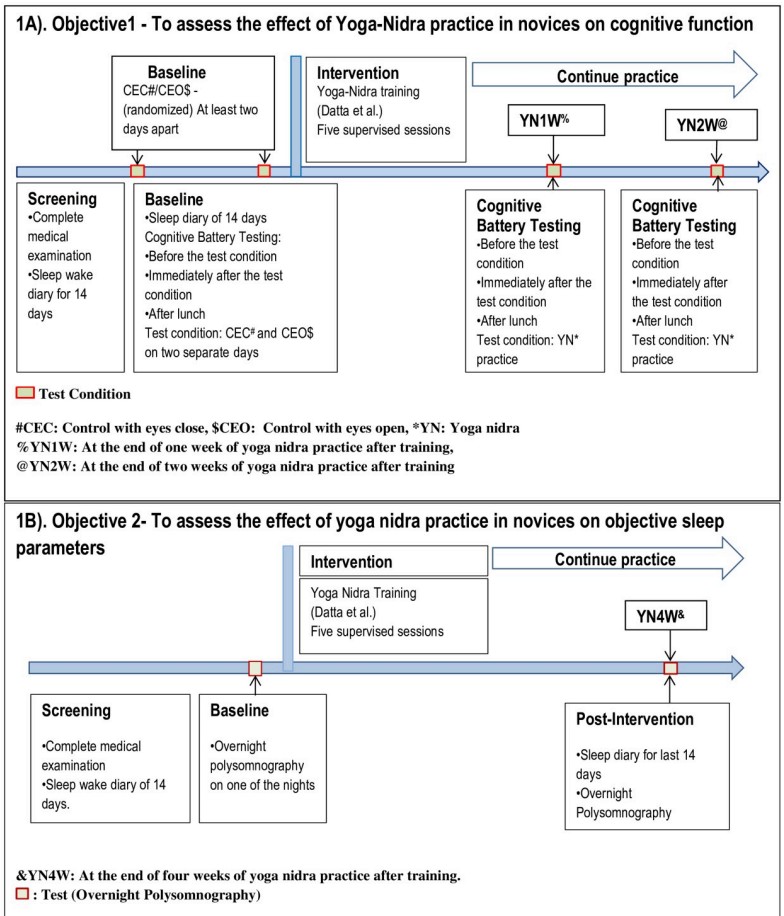

**Fig 1. Study Design for Effect of Yoga Nidra Practice on Cognitive Function (Fig 1A) and on Objective Sleep Parameters (Fig 1B).**

The study was conducted during 2021–2022 after institutional ethical clearance (IEC/2019/255 dated 25 Apr 2019). The study was registered in CTRI.in, vide no. CTRI/2021/031349. Local advertisements in the city were placed and volunteers assessed for eligibility. A minimum of 30 participants for each of the objectives were considered for this pilot study.

## 2.2 Screening of participants & enrollment

Healthy young male volunteers with a consistent sleep wake schedule were recruited. All participants underwent a thorough examination by a medical practitioner. Sleep wake schedule was monitored, and a baseline sleep diary of 14 days obtained. Exclusion criteria included any sleep disorder, depression, h/o psychiatric disorder, and h/o acute illness which were likely to affect sleep wake schedule. Women were excluded from this pilot study due to the confounding effect of the various menstrual phases on sleep [22, 23].

## 2.3 Intervention: Yoga nidra practice

A pre-recorded yoga nidra audio session was used (available *at* https://youtu.be/0fmMlELn-Ug). Yoga nidra was practiced for 20 minutes with three minutes of instructions before and

after the practice. The practice consisted of seven steps, namely: preparation, *samkalpa* (resolution), body part awareness, breath awareness, feeling and sensation, visualization, and ending of practice. Initially, yoga nidra training was done using supervised Yoga nidra sessions as per the therapeutic model developed earlier. During these supervised sessions, the investigator followed the 'guidelines for observer' developed earlier [8, 14]. These guidelines helped check the compliance and attentiveness of the subject during the session. This was done by looking for visual cues to ascertain that the participant followed the instructions of the audio. Participants having restlessness to an extent of affecting them to concentrate on the practice were asked to discontinue the session by the observer. At the end of the session, the participant was asked to review the audio and identify any portion that may have been missed during the session. This improved the effectiveness and completeness of yoga nidra practice on the subsequent days. After each supervised session any complaints of the participants or inability to do any part of the practice were addressed. The participants continued the practice on their own after the supervised training was done. Follow up testing on participants was done as per the study design given in Fig 1(A) and 1(B).

## 2.4 Sleep Diary (SD) analysis

All participants filled a baseline sleep diary for 14 days. The sleep diary was filled twice a day, once in the morning just after getting up and again at night just before bedtime. The participants continued to fill the sleep diary during the intervention. The sleep diary parameters obtained were: time in bed (sdTIB), total sleep time (sdTST), sleep onset latency (sdSOL), wake after sleep onset (sdWASO), sleep quality (sdSQ), and sleep efficiency (sdSE)[14]. These parameters were extracted from the baseline two-week data and also from the post-intervention two week data.

Data Analysis: Sleep diary data was checked for normality using Shapiro-Wilks test. If data was normally distributed, paired *t*-test was used to compare baseline sleep diary parameters with post intervention values. For non-normal data several transformations like square, square root, log, and exponential were tried. If normality was not achieved, the Wilcoxon signed rank test was used. Statistical significance was considered at alpha 0.05. All statistical analyses were done using R-software (version 4.1.2).

## 2.5 Methodology for assessing the effect of yoga nidra practice on cognitive function

The cognition test battery (CTB) was employed using the Joggle Research Platform (USA) [24]. Our CTB paradigm consisted of ten tests in a standard sequence of motor praxis task (MPT), visual object learning task (VOLT), fractal-2-back (NBACK), abstract matching (AIM), line orientation task (LOT), emotion recognition task (ERT), matrix reasoning task (MRT), digital symbol substitute task (DSST), balloon analog risk task (BART), and psycho-motor vigilance task (PVT:10min). The various cognition domains that were assessed by these 10 tests were: MPT–sensory motor speed, VOLT–spatial learning and memory, NBACK–working memory, AIM–abstraction and concept information, LOT–spatial orientation, ERT–emotion identification, MRT–abstract reasoning, DSST–complex scanning and visual tracking, BART–risk decision making, and PVT–vigilant attention [24].

CTB was performed on four test condition days which were control with eyes open (CEO), control with eyes close (CEC), at the end of 1st week of yoga nidra practice (YN1W), and at the end of 2nd week of yoga nidra practice (YN2W). All the participants completed the CTB tests three times on each test condition day i.e., at 1000h (First test of morning: Mr1), immediately after the test condition of CEC, CEO, yoga nidra practice at YN1W, or yoga nidra practice at

YN2W (Second test of morning: Mr2) and then in the afternoon at 1500h (post lunch: PL). All participants received similar food and working environment on each test condition day. Detailed representation of study design is shown in Fig 1(A).

CTB outcome variables studied were reaction time (for all 10 tests) and accuracy score for nine tests, except BART. For BART–'Adjusted Number of Pumps', 'Number of Explosions', 'Risk Propensity' and 'Cash collected' were analyzed [25]. 'Adjusted Pumps' is the average number of pumps on each balloon during BART when the balloon did not burst, 'Number of Explosions' is the total number of balloons exploded during BART, 'Risk Propensity' is the ratio of total pumps/balloons collected, and 'Cash collected' is the amount of reward earned in BART [24].

The standard accuracy outcome was 'proportion correct' ranging from 'zero' to 'one', where 'zero' and 'one' represented worst and best possible performance respectively. For the MPT, the distance from the center of the square to the point where participant touches was used as an accuracy score. The center of square translates to 'one' accuracy score and an edge of square to 'zero' accuracy score, with linear scaling between center and edges. For LOT, the accuracy score was calculated as 'three' minus the average number of clicks off, which was then divided by 'three'. For tests with average number of clicks off is more than 'three', the accuracy score was set to 'zero'. For PVT, the accuracy score was calculated as 'one'–[(number of lapses + number of false starts) / (number of stimuli + number of false starts)]. PVT- number of false starts was analyzed apart from reaction time and accuracy [24]. ERT was also studied for various types of stimuli apart from reaction time and accuracy.

Data Analysis: One-Way Repeated Measures (*RM*) Analysis of Variance (*ANOVA*) for checking significant change in each of the outcome variable between different test conditions (CEO, CEC, YN1W, and YN2W) and at different times (Mr1, Mr2, or PL) was used for normal data. If the data was not normal, Friedman's test was used. In instances where the sphericity assumption was not met, the reported *p*-values associated with the *F*-statistics were adjusted via the Greenhouse-Geisser correction. If the *ANOVA* model was significant, post-hoc analysis (pair wise comparison) was done using paired *t*-test with Bonferroni Correction. Wilcoxon Signed Rank Test with Bonferroni Correction was used for post-hoc analysis (pair wise comparison) if Friedman's Test was significant. Bonferroni Correction was used to reduce the probability of type-1 error. All reported *p*-values were two-tailed and statistical significance was assumed if *p*-value $< 0.05$. For *t*-test, effect size was calculated using Cohen's d (*d*) where $d < 0.5$ was small effect, $0.5 \leq d < 0.8$ was moderate effect, and $d \geq 0.8$ was large effect. For Wilcoxon Signed Rank Test, effect size was calculated using Wilcoxon effect size (*r*) where $r < 0.3$ was small effect, $0.3 \leq r < 0.5$ was moderate effect, and $r \geq 0.5$ was large effect. For ANOVA, generalized eta-squared ($\eta^2$) was used as effect size, where $\eta^2 < 0.06$ was small effect, $0.06 \leq \eta^2 < 0.14$ was medium effect, and $\eta^2 \geq 0.14$ was large effect.

All statistical analyses were done using R (version 4.1.2).

## 2.6 Power spectral density (PSD) of EEG daytime recording during CTB

The power spectral analysis of EEG data was performed using MNE-Python, an open-source library for visualizing, analyzing and exploring the raw EEG signal [26]. PSD analysis was planned out for the EEG data of 12 participants who volunteered for the EEG acquisition during CTB. This included loading data, pre-processing, and segmentation of the continual data on the obtained epoched data. Time-Frequency investigation was performed and finally spectral-parameters of the EEG signal was extracted [27]. Spectral investigation was based on frequency bands. PSD is the reflection of frequency components of EEG signal [28]. Preprocessing included low and high frequency filtering of the EEG signal, notch filtering and

removal of bad segments of the data (made by visual screening of data for sudden artifacts). Eye movements were repaired using the artifacts repair regression technique, custom re-referencing, and segmentation of the EEG data to epochs. Filtering was done at low-frequency (1Hz), high-frequency (90Hz) and notch-filter at 50Hz. After manual marking of artifacts and bad segments, MNE-Python library was used to automatically exclude annotated spans of data while creating the epochs from the continual EEG data. Special care was taken to avoid segments with artifacts or noise so that artifact free data would be extracted for further analysis. The EEG signal acquisition was based on the domain of time and future extraction. Slow eye movement artifacts in the EEG were repaired using computed coefficients [29].

Segmentation was performed on the clean raw EEG data to segment the EEG data to epochs. 'Feature extraction' was done based on 'Time-Frequency domain' [27, 30]. The time-frequency domain depicts the distribution of power of the EEG signal with respect to Time-Frequency plane. The computation of PSD was carried out using Welch's Technique. The decomposition of signal to frequency bands were done using Fourier transforms [27]. The ranges of the bands were delta (1-4Hz), theta1 (4-6Hz), theta2 (6-8Hz), alpha1 (8-10Hz), alpha2 (10–12.5Hz) and beta (13-30Hz) [6]. The features used in the EEG signal analysis were relative-band-power and ratios between the bands [31]. On each test segment spectral analysis was performed for all the EEG channels. Using 256Hz sampling rate of EEG, feature extraction was done from the desired test segments for all the EEG channels on every individual participant using 512ms window duration. The specified spectral analysis methodology was employed for all the participants on all test conditions i.e., CEC, CEO, YN1W, and YN2W for all the times i.e., Mr1, Mr2, and PL. PSD values were compared for various frequency bands and as ratios. The various parameters thus studied were PSD values of delta, theta1, theta2, alpha1, alpha2, beta and ratios of delta/beta, theta1/beta, theta2/beta, theta1/alpha1, theta2/alpha1, theta1/alpha2, theta2/alpha2, delta/theta1, delta/theta2, delta/alpha1, and delta/alpha2.

Data Analysis: One-way *ANOVA* model was used for checking significance of difference in the PSD outcome parameters and post-hoc analysis using Tukey post-hoc test was done, if the data was normal. Otherwise Kruskal Wallis test was used with Wilcoxon test for post-hoc analysis. Statistical significance was considered at alpha 0.05. All statistical analyses were done using R (version 4.1.2).

## 2.7 Methodology for assessing the effect of yoga nidra practice on objective sleep parameters

Overnight PSG was done in 30 participants to assess the effect of yoga nidra on objective sleep parameters. PSG was conducted according to AASM criteria [32] using 'SOMNOMEDICS©, Germany PSG System'. EEG, EOG, EMG channels were placed for scoring sleep-wake stages using DOMINO© software version 3.0.0.6. F3, F4, C3, C4, O1, and O2 EEG channels were placed as per the 10–20 system. PSG study was done during the night under standard conditions. The impedance was tried to be kept below 5KΩ. PSG was performed twice, at baseline (BL PSG) and post-intervention i.e., after four weeks of yoga nidra practice after initial training (PI PSG).

PSG data files were analyzed by KD in groups of 15–20 files and each was coded to blind KD from the identity state of subject data i.e., of baseline (BL) or post-intervention (PI). The files once analyzed were decoded by AB for further analysis.

Various parameters obtained using PSG were time in bed (TIB), total sleep time (TST), wake duration, wake after sleep onset (WASO), and duration of various stages of sleep (i.e., N1, N2, N3 and REM). Amplitude of alpha, beta, theta and delta waves was recorded in $\mu V^2$. Percentages for sleep, wake, REM, and Non-REM sleep were calculated and analyzed.

Data Analysis: Normality of data was checked using Shapiro-Wilk's test. Quantile-Quantile plot for each of the outcome parameter was made. If normality was not obtained even after square root, square, log, and exponential transformations then analysis was done using Wilcoxon Signed Rank Test. All statistical analyses were done using R (version 4.1.2).

## 3. Results and discussion

41 participants were enrolled in the study. Details of number of participants screened, allocated, and finally analyzed are shown in Fig 2. Three participants could not report for testing

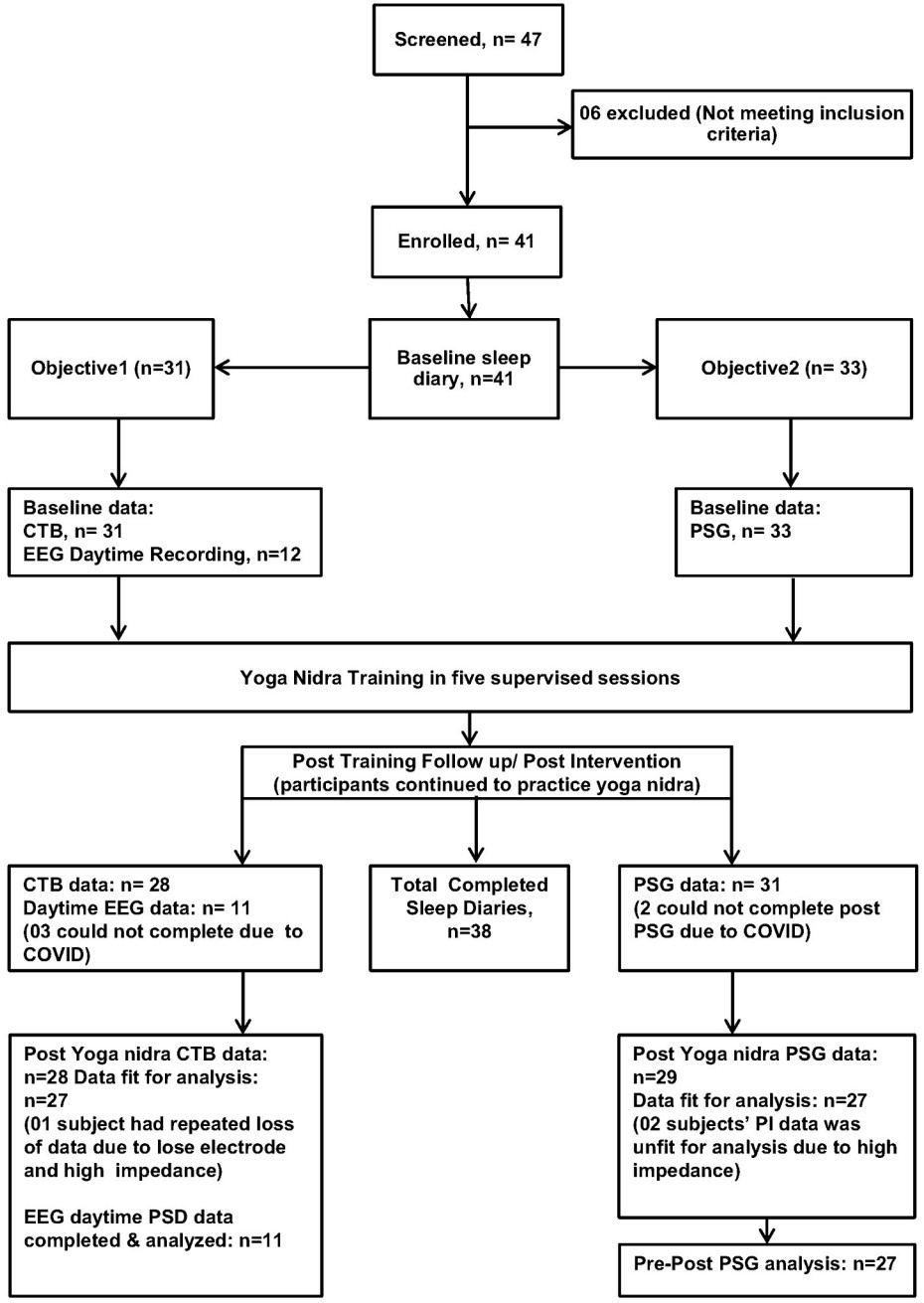

**Fig 2. Details of the participants screened and finally analyzed.**

due to the COVID pandemic during the time of study, though they continued the intervention. No adverse effects were reported. Those participants who were unable to report for either CTB or PSG were not included in the analysis as shown in Fig 2. The demographic profile of participants is provided in S1 Table. In four participants, sleep diary data for some days was missing in either the baseline or post-intervention data but was < 10%. This was imputed with the mean of rest data values. This was either in baseline (n = 3) or in post-intervention (n = 1) and none in both baseline and post-intervention. Analysis was done on these completed sleep diaries (n = 38). Sleep diary analysis is shown in Table 1. We found a significant improvement in total sleep time, sleep quality, sleep onset latency, and WASO which was reported subjectively by the patients using sleep diary.

Polysomnography data (baseline; post-intervention; $p$-value, effect size) of TIB (428·09 ±82·13; 423·02±72·96; 0·67, $d = 0.09$), TST (372·36±68·46; 387·70±74·94; 0·24, $d = 0.24$), and of Total Non-REM (340·73±111·95; 341·88±73·48; 0·95, $d = 0.01$) was not significant. Pre-post-intervention analysis of other PSG parameters is shown in Fig 3. Our results showed an objective improvement in nighttime sleep after yoga nidra practice in novices. We found an increase in sleep efficiency, improvement in WASO and a significant improvement in delta sleep (%) of slow wave sleep. It is likely that a reduction in sympathetic drive and an increase in parasympathetic drive occurred due to the practice of yoga nidra during the morning hours, resulting in improved slow wave sleep later in the night [33]. However, the exact mechanism is still not elucidated. The objective improvement of the delta sleep in slow wave sleep has been an important factor in improvement of sleep quality. Enhancement of slow wave sleep is reported using auditory stimuli during sleep, gaboxadol, tiagabine and transcranial direct cranial stimulation [34–36]. Slow wave sleep enhancement has been found to improve attention, learning, memory and performance [35–37]. Slow waves during sleep have been directly or indirectly linked to synaptic strength as a part of synaptic homeostasis theory [38].

Cognition testing battery results for test condition wise and testing time wise comparisons in reaction time are shown in Figs 4 and 5 respectively. The comparisons of condition wise and testing time in accuracy scores are shown in S1 and S2 Figs respectively. Master sheet of CTB results in *Mean (SD)* is given in S2 Table. BART results for 'adjusted number of pumps', 'risk propensity', 'cash collected', and 'number of balloons burst' are shown in S3 and S4 Figs for test condition and testing time wise comparisons respectively. PVT result of 'number of false starts' is shown in S5 Fig.

**Table 1. Sleep diary parameters of participants before (baseline) and after four weeks of Yoga nidra practice (post intervention) after training.**

| Variables | Baseline | | Post-intervention | | Wilcoxon test (P-value) | Effect size (r) | Difference: PI-BL | | 95% CI (PI-BL) | | %Change = (PI-BL)/BL*100 |
|---|---|---|---|---|---|---|---|---|---|---|---|
| | Mean | SD | Mean | SD | | | Mean | SD | LL | UL | |
| sdTIB(min) | 409.39 | 80.28 | 427.51 | 66.96 | <0.001 | 0.24 | 18.12 | 98.99 | 9.69 | 26.55 | 4.43% |
| sdTST(min) | 383.95 | 80.27 | 408.96 | 69.59 | <0.001 | 0.32 | 25.01 | 100.72 | 16.44 | 33.59 | 6.52% |
| sdSOL(min) | 21.75 | 21.05 | 16.31 | 17.90 | <0.001 | 0.25 | -5.45 | 25.32 | -7.60 | -3.29 | -25.03% |
| sdWASO(min) | 3.99 | 9.93 | 2.29 | 5.72 | <0.001 | 0.18 | -1.70 | 10.69 | -2.61 | -0.79 | -42.59% |
| sdSQ | 7.94 | 1.29 | 8.25 | 1.16 | <0.001 | 0.21 | 0.31 | 1.49 | 0.18 | 0.43 | 3.84% |
| sdSE (%) | 93.75 | 5.61 | 95.57 | 4.63 | <0.001 | 0.32 | 1.82 | 6.70 | 1.25 | 2.39 | 1.94% |

*Note*. Wilcoxon signed rank test was used to compare baseline with post-intervention and effect size was calculated using Wilcoxon effect siI($r$) where $r < 0.3$ (small effect), $0.3 \leq r < 0.5$ (moderate effect), and $r \geq 0.5$ (large effect).

sdTIB(min): Time in bed in minutes, sdTST(min): Total sleep time in minutes, sdSOL(min): Sleep onset latency in minutes, sdWASO(min): Wake after sleep onset in minutes, sdSQ: Sleep quality (subjective rating between 0–10), sdSE(%): Sleep efficiency calculated as (TIB/TST)*100, PI: Post intervention reading, BL: Baseline reading, CI: Confidence interval, LL: Lower limit, UL: Upper limit.

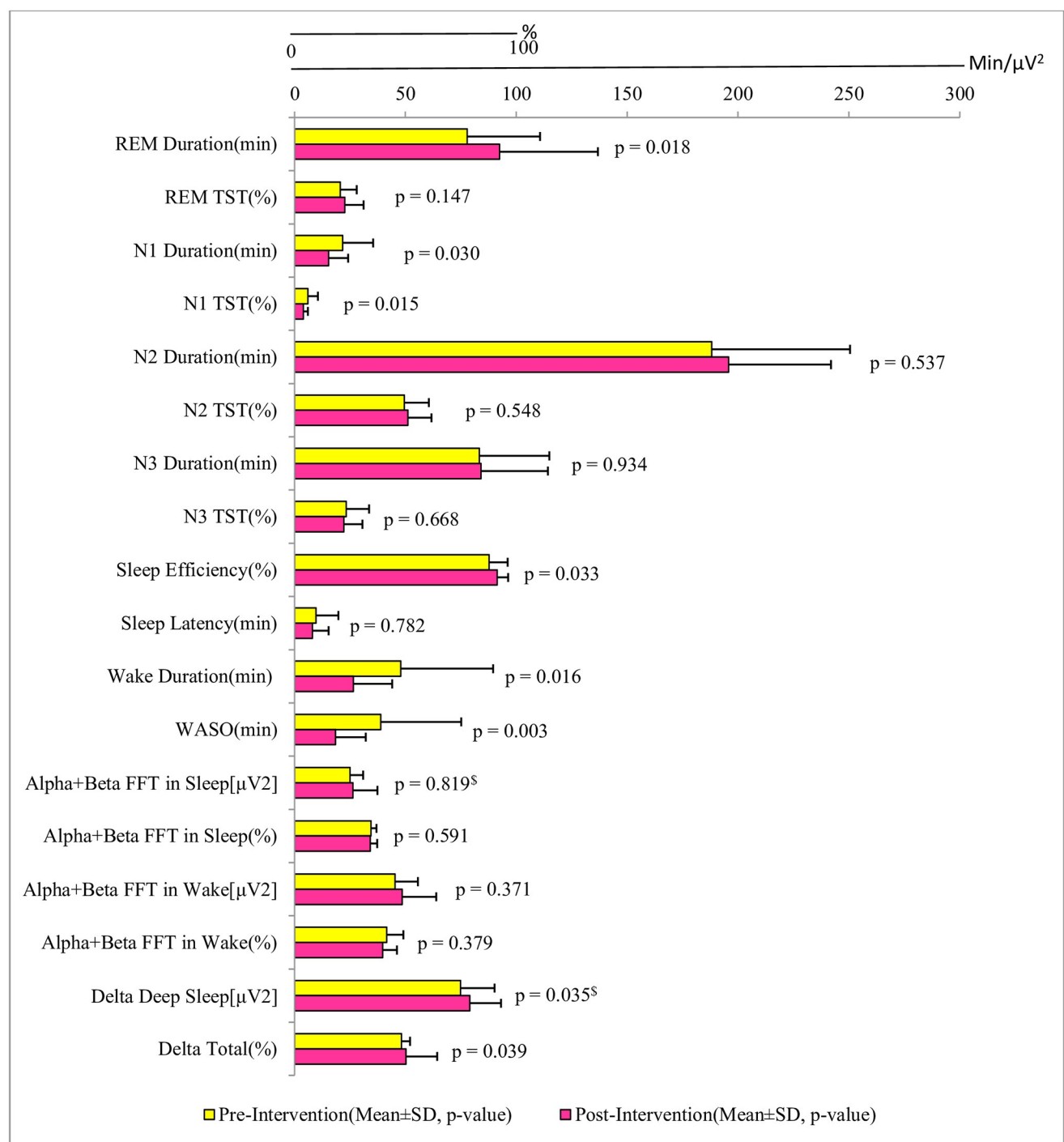

**Fig 3. Comparison of objective sleep parameters using polysomnography in participants before (baseline) and after (post intervention) Yoga Nidra Practice.** (Note- $: *p*-values computed using Wilcoxon signed rank test. Rest all *p*-values are computed using two-tailed paired *t*-test).

Yoga nidra practice resulted in significant improvement in reaction time with no deterioration in accuracy of all cognitive battery tests i.e., MPT, VOLT, NBACK, AIM, LOT, ERT, MRT, DSST, BART, and PVT. This implies an increase in processing speed. Certain tests like AIM, ERT, NBACK, and VOLT actually showed an increase in accuracy too. This is possible

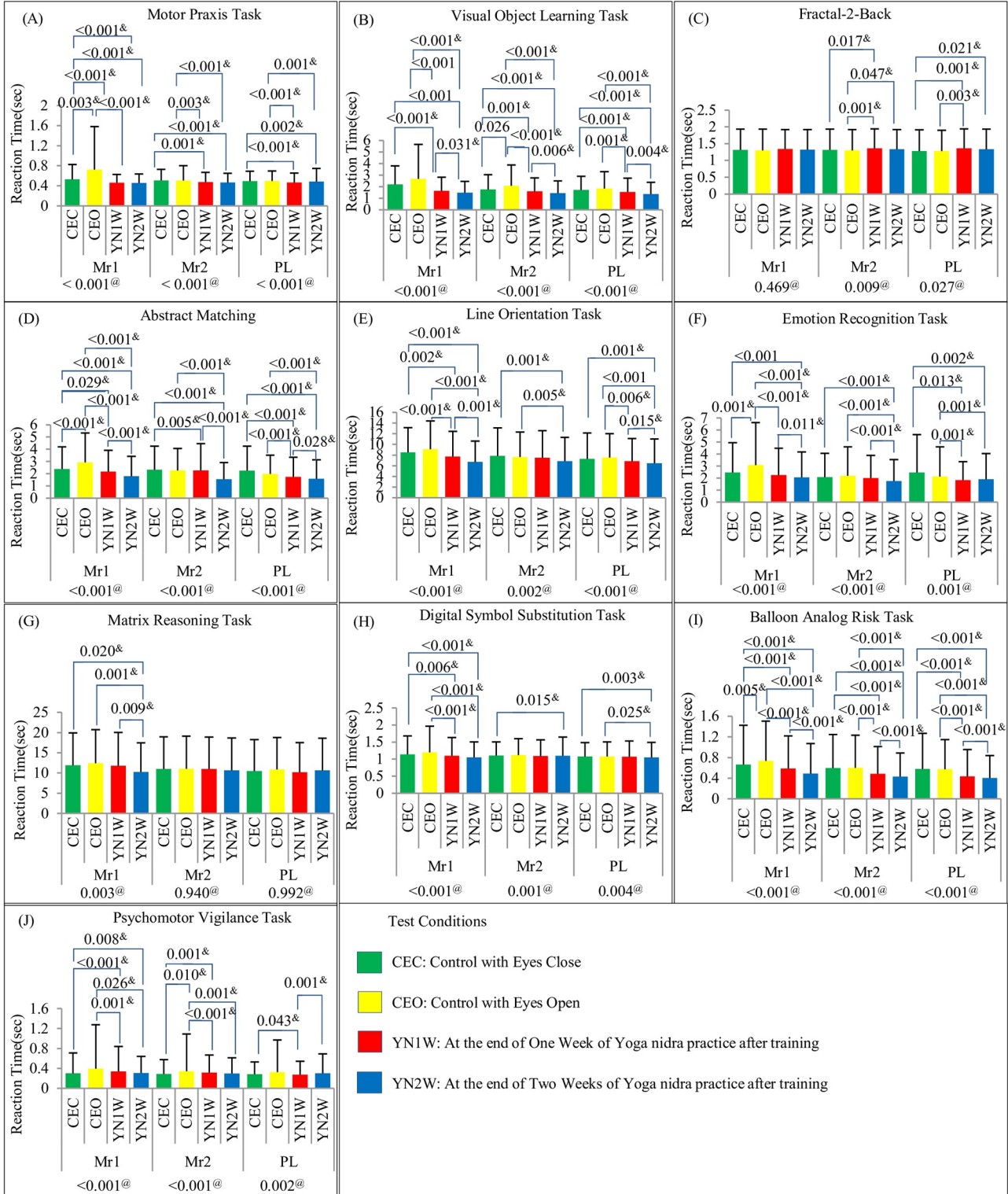

**Fig 4. Cognitive Testing Battery Results Showing Reaction Time (Mean (SD)) in (A)MPT, (B)VOLT, (C)NBACK, (DIM, (E)LOT, (F)ERT, (G) MRT, (H)DSST, (I)BART, and (J)PVT with Test Condition Wise Comparison.** (Comparison model for test condition i.e. CEC, CEO, YN1W, and YN2W is shown below each time i.e. Mr1, Mr2, and PL. Significant post-hoc p-values are also depicted). #: One-Way Repeated Measure (RM) Analysis of Variance (ANOVA) p-values $: Post-hoc analysis (pairwise comparison) is done using paired t-test with Bonferroni correction (significant changes indicated by p-values given above the bars). @: Friedman's test p-value (if data is not normally distributed) &: Post-hoc analysis (pairwise comparison)

is done using Wilcoxon Signed-Rank test with Bonferroni correction (significant changes indicated by p-values given above the bars). Mr1: Test just before CEC, CEO, YN1W or YN2W; Mr2: Tests conducted just after CEO, C EC, YN1W or YN2W; PL: Test conducted after the lunch.

due to the effects of the practice either directly or indirectly on the various brain regions involved in performing these tests. It is also possible that these effects are due to the local sleep seen in central, occipital and parietal areas during the rotation of consciousness in yoga nidra practice as brought out by Datta et al in a previous study on novices [6]. Following local sleep, task performance is found to be enhanced [39]. Though not examined in this study, but probably the increase in synaptic strength as per the synaptic hypothesis might also be responsible for these findings [38].

Accuracy improved in Abstract Matching test which is specifically related to prefrontal cortex and includes abstraction and concept formation. This accuracy was found to be increased in early morning which may be as a direct or indirect effect related to improved sleep after yoga nidra practice. Slow wave sleep is essential for enhanced prefrontal cortex activity. Working memory as shown with NBACK test also showed significant improvement with yoga nidra practice. The actual number of 'adjusted number of pumps' on balloons collected as cash, showed a significant increase in post lunch session after two weeks of practice. This meant that the number of bursts on the balloons only which were encashed was higher. ERT showed a significant deterioration in the early morning task scores, though, immediately after yoga nidra practice, it did not show any significant change in scores as compared to baseline. This was further studied by analyzing the type of stimuli where there was a change. ERT analysis for various types of stimuli is shown in S3 Table. On further analysis, it was evident that yoga nidra intervention only showed deterioration in neutral stimuli recognition without affecting the emotion recognition of happy stimuli and in fact improved recognition of anger and fear stimuli. The exact mechanism of this cannot be understood and would require further study. The relative improvement in non-neutral emotional recognition task is also interesting and requires more deliberation with special emphasis on emotion recognition and thus future study may aim at the effects of yoga nidra practice on social interaction.

12 participants volunteered for daytime EEG recording during CTB. To understand the mechanism of the effect of the yoga nidra practice during these cognitive tasks EEG PSD values at O1, F3, and C3 were analyzed. Summary of significant test results in EEG analysis of power spectra density using Python are shown in Table 2. These showed significant changes in VOLT, in which accuracy of CTB was also found to be increased. Significant increase in PSD in delta and theta frequencies was seen at O1 location after two weeks of yoga nidra practice. At C3 location also, PSD of theta remained significantly increased, immediately after yoga nidra practice during VOLT. A local sleep type phenomenon with some direct or indirect effects on areas associated with VOLT i.e. middle-temporal cortex and hippocampus, might explain this positive effect on spatial learning in memory. In our study, it is also interesting to note that after the practice also, delta propensity continued without reduction in accuracy, rather improvement in reaction time. It can only be postulated that increased PSD of delta waves implying enhancement of slow waves may be due to the increased strength of synapses [40] as highlighted by Tononi et al. Increased delta propensity during tasks on EEG PSD values, especially in the early morning readings after two weeks of yoga nidra practice, hints also at a not so immediate effect of practice, and might be a post good night sleep effect after practicing yoga nidra for two weeks in the mornings.

The study highlights the important effect of yoga nidra practice on cognitive processing. However, this study demonstrates the effect of only two weeks of yoga nidra practice. Another

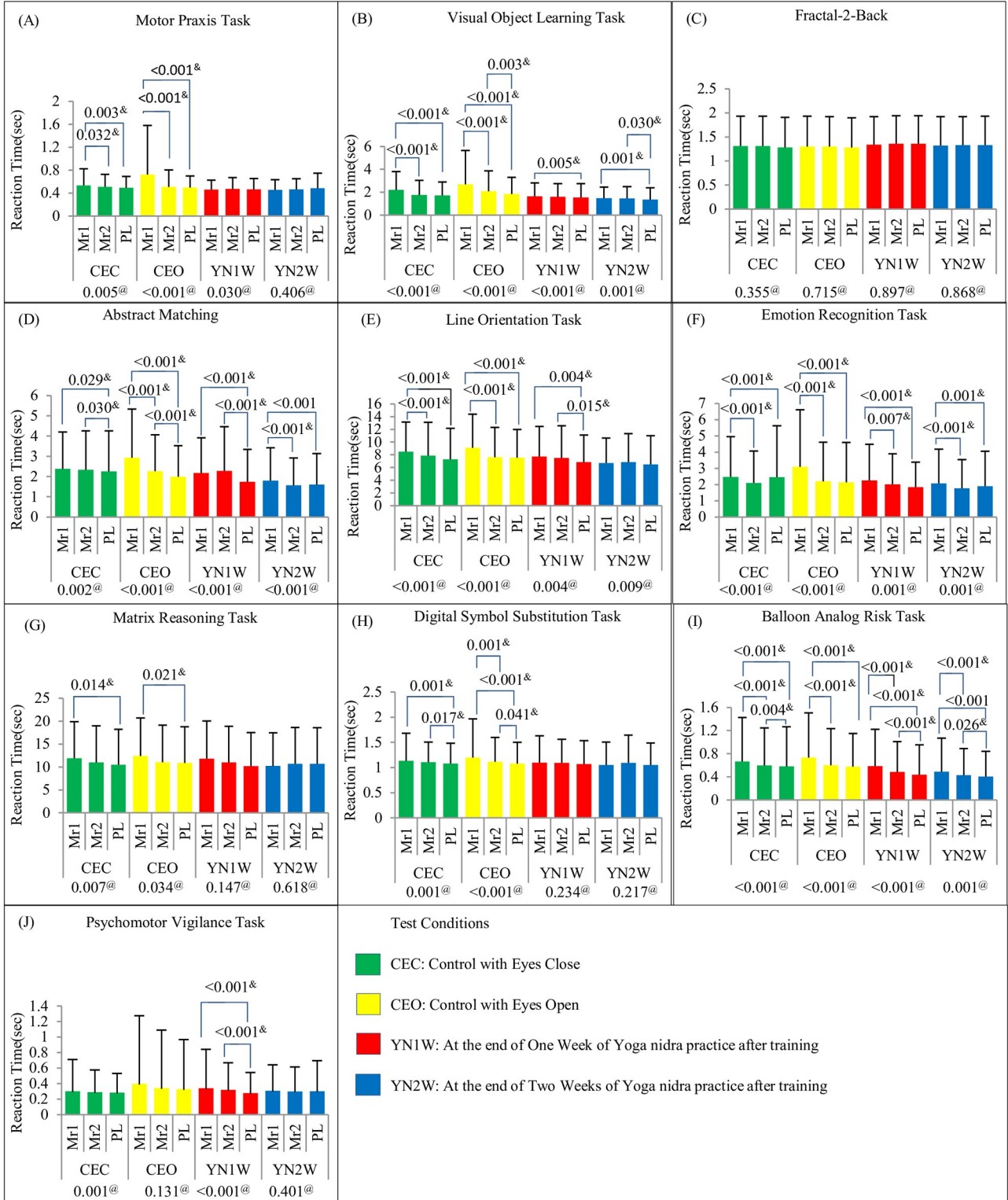

**Fig 5. Cognitive Testing Battery Results Showing Reaction Time (Mean (SD)) in (A)MPT, (B)VOLT, (C)NBACK,I)AIM, (E)LOT, (F)ERT, (G) MRT, (H)DSST, (I)BART, and (J)PVT with Time Wise Comparison.** (Comparison model for time i.e. Mr1, Mr2, and PL is shown below each test condition i.e. CEC, CEO, YN1W, and YN2W. Significant post-hoc p-values are also depicted)#: One-Way Repeated Measure (RM) Analysis of Variance (ANOVA) p-values $: Post-hoc analysis (pairwise comparison) is done using paired t-test with Bonferroni correction (significant changes indicated by p-values given above the bars). @: Friedman's test p-value (if data is not normally distributed) &: Post-hoc analysis (pairwise comparison) is done using

Wilcoxon Signed-Rank test with Bonferroni correction (significant changes indicated by p-values given above the bars). Mr1: Test just before CEC, CEO, YN1W or YN2W; Mr2: Tests conducted just after CEO, C EC, YN1W or YN2W; PL: Test conducted after the lunch.

**Table 2. EEG PSD significant results in frequency bands during CTB at various EEG locations.**

| CBT Task | EEG channel | Test time | Frequency band | ANOVA (*p*-value) | Effect size ($\eta^2$) | Post-hoc analysis test (only significant results) | | | Difference* | 95% CI for Difference* | | % Change |
|---|---|---|---|---|---|---|---|---|---|---|---|---|
| | | | | | | Test condition | Mean | SD | (p-value) | | L | U | |
| VOLT | O1 | Mr1 | delta/alpha1 | 0.04 | 0.24 | CEC | 5.80 | 4.57 | 0.03 | -3.25 | -6.21 | -0.29 | -56.08% |
| | | | | | | CEO | 2.55 | 1.08 | | | | | |
| | | | delta/alpha2 | 0.04 | 0.24 | CEC | 6.55 | 5.05 | 0.03 | -3.77 | -7.19 | -0.36 | -57.58% |
| | | | | | | CEO | 2.77 | 1.15 | | | | | |
| | | | delta/theta2 | 0.03 | 0.26 | CEC | 4.20 | 2.94 | 0.03 | -2.06 | -3.97 | -0.15 | -49.02% |
| | | | | | | CEO | 2.14 | 0.78 | | | | | |
| | | | | | | CEC | 4.20 | 2.94 | 0.04 | -2.12 | -4.18 | -0.05 | -50.45% |
| | | | | | | YN2W | 2.08 | 0.50 | | | | | |
| | | | Theta1(4-6Hz) | 0.01 | 0.33 | CEO | 0.12 | 0.02 | 0.01 | 0.04 | 0.01 | 0.07 | 33.05% |
| | | | | | | YN2W | 0.16 | 0.03 | | | | | |
| | | | | | | YN1W | 0.11 | 0.02 | 0.01 | 0.05 | 0.01 | 0.08 | 39.82% |
| | | | | | | YN2W | 0.16 | 0.03 | | | | | |
| | | | Theta2(6-8Hz) | 0.02 | 0.29 | CEC | 0.07 | 0.03 | 0.03 | 0.04 | 0.00 | 0.07 | 55.07% |
| | | | | | | YN2W | 0.11 | 0.02 | | | | | |
| | | | | | | YN1W | 0.07 | 0.02 | 0.05 | 0.03 | 0.00 | 0.07 | 44.60% |
| | | | | | | YN2W | 0.11 | 0.02 | | | | | |
| | | | theta2/alpha2 | 0.02 | 0.28 | CEO | 1.33 | 0.44 | 0.02 | 0.73 | 0.08 | 1.38 | 54.78% |
| | | | | | | YN2W | 2.06 | 0.80 | | | | | |
| | | PL | Theta2(6-8Hz) | 0.03 | 0.30 | CEO | 0.08 | 0.02 | 0.02 | 0.03 | 0.00 | 0.05 | 37.33% |
| | | | | | | YN2W | 0.10 | 0.02 | | | | | |
| | C3 | Mr2 | Theta2(6-8Hz) | 0.04 | 0.26 | CEC | 0.06 | 0.02 | 0.04 | 0.04 | 0.00 | 0.08 | 72.73% |
| | | | | | | YN1W | 0.10 | 0.04 | | | | | |
| BART | O1 | Mr2 | delta/beta | 0.03 | 0.27 | CEC | 0.95 | 0.46 | 0.04 | 1.11 | 0.04 | 2.18 | 116.72% |
| | | | | | | YN2W | 2.06 | 1.28 | | | | | |
| | | | | | | CEO | 0.95 | 0.50 | 0.03 | 1.11 | 0.09 | 2.13 | 117.34% |
| | | | | | | YN2W | 2.06 | 1.28 | | | | | |
| PVT | O1 | Mr2 | Theta2(6-8Hz) | 0.02 | 0.28 | CEO | 0.07 | 0.02 | 0.04 | 0.03 | 0.00 | 0.05 | 38.03% |
| | | | | | | YN1W | 0.10 | 0.03 | | | | | |
| ERT | F3 | PL | delta/alpha1 | 0.02 | 0.34 | CEC | 4.51 | 1.48 | 0.03 | 5.75 | 0.57 | 10.90 | 127.61% |
| | | | | | | YN1W | 10.26 | 5.90 | | | | | |
| | | | delta/alpha2 | 0.02 | 0.34 | CEC | 6.41 | 2.64 | 0.04 | 8.93 | 0.48 | 17.40 | 139.25% |
| | | | | | | YN1W | 15.35 | 10.09 | | | | | |
| | | | | | | CEO | 6.61 | 3.54 | 0.03 | 8.73 | 0.53 | 16.90 | 132.01% |
| | | | | | | YN1W | 15.35 | 10.09 | | | | | |
| | | | delta/theta1 | 0.03 | 0.32 | CEC | 1.74 | 0.22 | 0.03 | 0.49 | 0.04 | 0.93 | 28.04% |
| | | | | | | YN1W | 2.22 | 0.39 | | | | | |
| | | | delta/theta2 | 0.03 | 0.32 | CEC | 3.08 | 0.64 | 0.03 | 2.27 | 0.17 | 4.37 | 73.75% |
| | | | | | | YN1W | 5.35 | 2.18 | | | | | |
| | | | Theta1/alpha2 | 0.03 | 0.31 | CEO | 3.44 | 1.42 | 0.04 | 3.31 | 0.09 | 6.53 | 96.22% |
| | | | | | | YN1W | 6.75 | 3.81 | | | | | |

(*Continued*)

**Table 2.** (Continued)

| CBT Task | EEG channel | Test time | Frequency band | ANOVA (*p*-value) | Effect size ($\eta^2$) | Post-hoc analysis test (only significant results) | | | | Difference* | 95% CI for Difference* | | % Change |
|---|---|---|---|---|---|---|---|---|---|---|---|---|---|
| | | | | | | Test condition | Mean | SD | (p-value) | | L | U | |
| LOT | F3 | PL | Delta/theta1 | 0.03 | 0.31 | CEC | 1.82 | 0.46 | 0.05 | 0.53 | 0.00 | 1.05 | 29.14% |
| | | | | | | YN1W | 2.35 | 0.23 | | | | | |
| MPT | C3 | Mr2 | Theta1(4-6Hz) | 0.03 | 0.29 | CEC | 0.07 | 0.03 | 0.02 | 0.05 | 0.01 | 0.09 | 75.39% |
| | | | | | | YN1W | 0.11 | 0.02 | | | | | |
| MRT | C3 | Mr1 | Delta/beta | 0.04$^\$$ | 0.20 | CEC | 1.67 | 0.37 | 0.05$^\$$ | -0.74 | -1.29 | -0.18 | -44.16% |
| | | | | | | CEO | 0.93 | 0.42 | | | | | |

*Note.* One-way ANOVA model was used for checking significant change and post-hoc analysis is done using Tukey post hoc test except at $ where Kruskal Wallis test was used and post-hoc analysis is done using Wilcoxon test. Effect size was calculated using generalized eta-squared ($\eta^2$), where $\eta^2 < 0.06$ (small effect), $0.06 \leq \eta^2 < 0.14$ (medium effect), and $\eta^2 \geq 0.14$ (large effect).VOLT: Visual Object Learning Task, BART: Balloon Analog Risk Task, PVT: Psychomotor Vigilance Task, ERT: Emotion Recognition Task, LOT: Line Orientation Task, MPT: Motor Praxis Task, MRT: Matrix Reasoning Task, CEO: Control with Eyes Open, CEC: Control with Eyes Close, YN1W: At the end of one week of Yoga nidra practice after training, YN2W: At the end of two weeks of Yoga nidra practice after training·, Mr1: Test just before CEC, CEO, YN1W or YN2W, Mr2: Tests conducted just after CEO, CEC, YN1W or YN2W, PL: Tests conducted after the lunch.

*Difference is calculated as difference between two PSD parameters of two test conditions given in 'Test Condition' column, CI: Confidence interval, L: Lower limit, U: Upper limit.

drawback is the limited number of EEG locations used for analysis of cognitive tests. This pilot study was done using a pre-post intervention design to analyze the effects of this practice in novices and no active control groups were taken in this study. Though this study discusses the effect of practice as a pre-post intervention design, but studies with a larger sample, active control groups, longer exposure to yoga nidra, several EEG locations, and a randomized controlled trial may give a better perspective in analyzing the effects of this practice on cognitive processing. This study opens up an opportunity for the use of an easy-to-do practice of yoga nidra for population health using standardized supervised model [8,14, 41]. The study highlights the possible role of this practice in improving sleep and in promoting learning and memory amongst healthy participants. It might hold promise for patients with mild learning disability and mild cognition deterioration and possibly in its prevention specially in the aging population [42]. In the post-pandemic times, sleep is commonly found to be affected, which creates a risk of multiple disorders including neuropsychiatric disorders [43]. Planning large population based studies of different cultures [5] may also help assess its effects on insomnia globally. It has an immense role in improving sleep of the population at large and might be a way ahead for improving productivity at workplace [44]. An increased awareness of sleep problems and their management amongst primary healthcare physicians is vital [45]. Yoga nidra practice using formulated guidelines might ensure wellbeing of the society as it emerges from the effects of pandemic [41].

## Supporting information

**S1 Fig. Cognitive Testing Battery Results Showing Accuracy Measure (Mean (SD)) in (A) MPT, (B)VOLT, (C)NBACK, (D)AIM, (E)LOT, (F)ERT, (G)MRT, (H)DSST, and (I)PVT with Test Condition Wise Comparison.** (Comparison model for each test condition i.e. CEC, CEO, YN1W, and YN2W is shown below each time i.e. Mr1, Mr2, and PL. Significant post-hoc p-values are also depicted) #: One-Way Repeated Measure (RM) Analysis of Variance (ANOVA) p-values $: Post-hoc analysis (pairwise comparison) is done using paired t-test with Bonferroni correction (significant changes indicated by p-values given above the bars). @:

Friedman's test p-value (if data is not normally distributed) &: Post-hoc analysis (pairwise comparison) is done using Wilcoxon Signed-Rank test with Bonferroni correction (significant changes indicated by p-values given above the bars). Mr1: Test just before CEC, CEO, YN1W or YN2W Mr2: Tests conducted just after CEO, C EC, YN1W or YN2W PL: Test conducted after the lunch CEO: Control with Eyes Open CEC: Control with Eyes Close YN1W: At the end of one week of Yoga Nidra practice after training YN2W: At the end of two weeks of Yoga Nidra practice after training.
(TIF)

**S2 Fig. Cognitive Testing Battery Results Showing Accuracy Measure (Mean (SD)) in (A) MPT, (B)VOLT, (C)NBACK, (D)AIM, ILOT, (F)ERT, (G)MRT, (H)DSST, and (I)PVT with Time Wise Comparison.** (Comparison model for each test condition i.e. CEC, CEO, YN1W, and YN2W is shown below each time i.e. Mr1, Mr2, and PL. Significant post-hoc p-values are also depicted) #: One-Way Repeated Measure (RM) Analysis of Variance (ANOVA) p-values $: Post-hoc analysis (pairwise comparison) is done using paired t-test with Bonferroni correction (significant changes indicated by p-values given above the bars). @: Friedman's test p-value (if data is not normally distributed) &: Post-hoc analysis (pairwise comparison) is done using Wilcoxon Signed-Rank test with Bonferroni correction (significant changes indicated by p-values given above the bars). Mr1: Test just before CEC, CEO, YN1W or YN2W Mr2: Tests conducted just after CEO, C EC, YN1W or YN2W PL: Test conducted after the lunch CEO: Control with Eyes Open CEC: Control with Eyes Close YN1W: At the end of one week of Yoga Nidra practice after training YN2W: At the end of two weeks of Yoga Nidra practice after training.
(TIF)

**S3 Fig. BART results (Mean (SD)) of A) Adjusted Number of Pumps, B) Total cash collected ($), C) Risk Propensity, and D) Number of Balloons Burst Before (baseline) and After (post-intervention) Yoga Nidra Practice Test Condition Wise.** (P-values for time wise and test condition wise comparison model is shown below i.e. for Mr1, Mr2, and PL; and CEC, CEO, YN1W, and YN2W respectively. Significant post-hoc p-values are also depicted) Notes- #: One-Way Repeated Measure (RM) Analysis of Variance (ANOVA) p-values, $: Post-hoc analysis (pairwise comparison) is done using paired t-test with Bonferroni correction (significant changes indicated by p-values given above the bars)., @: Friedman's test p-value (if data is not normally distributed), &: Post-hoc analysis (pairwise comparison) is done using Wilcoxon Signed-Rank test with Bonferroni correction (significant changes indicated by p-values given above the bars). Mr1: Test just before CEC, CEO, YN1W or YN2W Mr2: Tests conducted just after CEO, C EC, YN1W or YN2W PL: Test conducted after the lunch CEO: Control with Eyes Open CEC: Control with Eyes Close YN1W: At the end of one week of Yoga Nidra practice after training YN2W: At the end of two weeks of Yoga Nidra practice after training.
(TIF)

**S4 Fig. BART results (Mean (SD)) of A) Adjusted Number of Pumps, B) Total cash collected ($), C) Risk Propensity, and D) Number of Balloons Burst Time Wise Before (baseline) and After (post-intervention) Yoga Nidra Practice.** (P-values for time wise and test condition wise comparison model is shown below i.e., for Mr1, Mr2, and PL; and CEC, CEO, YN1W, and YN2W respectively. Significant post-hoc p-values are also depicted). Notes- #: One-Way Repeated Measure (RM) Analysis of Variance (ANOVA) p-values, $: Post-hoc analysis (pairwise comparison) is done using paired t-test with Bonferroni correction (significant changes indicated by p-values given above the bars)., @: Friedman's test p-value (if data is not

normally distributed), &: Post-hoc analysis (pairwise comparison) is done using Wilcoxon Signed-Rank test with Bonferroni correction (significant changes indicated by p-values given above the bars). Mr1: Test just before CEC, CEO, YN1W or YN2W Mr2: Tests conducted just after CEO, C EC, YN1W or YN2W. PL: Test conducted after the lunch CEO: Control with Eyes Open. CEC: Control with Eyes Close YN1W: At the end of one week of Yoga Nidra practice after training. YN2W: At the end of two weeks of Yoga Nidra practice after training.
(TIF)

**S5 Fig. PVT-Number of False Starts (Mean (SD)) of Participants Before (baseline) and After (post-intervention) Yoga Nidra Practice Comparison in Test Condition wise (6(A)) and Time wise (6(B)).** Comparison model for each test condition i.e. CEC, CEO, YN1W, and YN2W is shown below each time i.e. Mr1, Mr2, and PL. Significant post-hoc p-values are also depicted.
(TIF)

**S1 File.**
(ZIP)

**S1 Table. Demographic profile of participants.**
(XLSX)

**S2 Table. Cognitive Testing Battery (CTB) parameters (Mean (SD)) of all tests in the participants.**
(DOCX)

**S3 Table. Emotion Recognition Task (ERT) Emotion-wise Analysis of Accuracy.**
(DOCX)

## Acknowledgments

Authors thank the study participants for their time and the lab staff for their support in conducting the study. Authors also thank Dr CV Apte for reviewing the manuscript.

## Author Contributions

**Conceptualization:** Karuna Datta, Rajagopal Srinath, Madhuri Kanitkar.

**Data curation:** Karuna Datta, Anna Bhutambare, Mamatha V. L., Yogita Narawa.

**Formal analysis:** Karuna Datta, Anna Bhutambare, Mamatha V. L., Yogita Narawa.

**Funding acquisition:** Karuna Datta, Madhuri Kanitkar.

**Investigation:** Rajagopal Srinath, Madhuri Kanitkar.

**Methodology:** Karuna Datta, Rajagopal Srinath, Madhuri Kanitkar.

**Project administration:** Karuna Datta.

**Resources:** Karuna Datta, Yogita Narawa, Rajagopal Srinath, Madhuri Kanitkar.

**Software:** Karuna Datta, Anna Bhutambare, Mamatha V. L.

**Supervision:** Karuna Datta, Yogita Narawa, Rajagopal Srinath, Madhuri Kanitkar.

**Validation:** Karuna Datta, Anna Bhutambare, Mamatha V. L., Yogita Narawa, Rajagopal Srinath.

**Visualization:** Anna Bhutambare, Mamatha V. L., Yogita Narawa.

**Writing – original draft:** Karuna Datta, Anna Bhutambare, Mamatha V. L.

**Writing – review & editing:** Karuna Datta, Rajagopal Srinath, Madhuri Kanitkar.

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
