## [Decision Letter · Decision Letter 0]

3 Sep 2023

PONE-D-23-21201Improved Sleep, Cognitive Processing and Enhanced Learning and Memory Task Accuracy with Yoga Nidra Practice in NovicesPLOS ONE

Dear Dr. Datta,

Thank you for submitting your manuscript to PLOS ONE. After careful consideration, we feel that it has merit but does not fully meet PLOS ONE’s publication criteria as it currently stands. Therefore, we invite you to submit a revised version of the manuscript that addresses the points raised during the review process.

As you can see we have received rather mixed reviews: Reviewer 1 recommends relatively minor revisions while Reviewers 2 and 3 recommend major revisions with respect to the methodological clarity and English language editing but are encouraging at the same time. Please submit your revised manuscript by Oct 18 2023 11:59PM. If you will need more time than this to complete your revisions, please reply to this message or contact the journal office at plosone@plos.org. Please include the following items when submitting your revised manuscript:A rebuttal letter that responds to each point raised by the academic editor and reviewer(s). You should upload this letter as a separate file labeled 'Response to Reviewers'.A marked-up copy of your manuscript that highlights changes made to the original version. You should upload this as a separate file labeled 'Revised Manuscript with Track Changes'.An unmarked version of your revised paper without tracked changes. You should upload this as a separate file labeled 'Manuscript'.

We look forward to receiving your revised manuscript.

One or more of the reviewers has recommended that you cite specific previously published works. Members of the editorial team have determined that the works referenced are not directly related to the submitted manuscript. As such, please note that it is not necessary or expected to cite the works requested by the reviewer.

Kind regards,

Kallol Kumar Bhattacharyya, MBBS MA PhD

Academic Editor

PLOS ONE

“Authors acknowledge Department of Science and Technology, Government of India for financial support vide Reference No. DST Satyam 2018/457 dated 28 May 2020 under Science and Technology for Yoga and Meditation (SATYAM) to carry out this work. Authors thank Dr CV Apte for reviewing the manuscript and thank the participants and staff for the support in conducting the study.”

“KD received the funding from Department of Science and Technology under Science and Technology for Yoga and Meditation (SATYAM), India available at https://dst.gov.in/, and was received vide their sanction order DST Satyam 2018/457 dated 28 May 2020.The funders had no role in study design, data collection and analysis, decision to publish, or preparation of the manuscript.”

Reviewers' comments:

Reviewer's Responses to Questions

**Comments to the Author**

1. Is the manuscript technically sound, and do the data support the conclusions?

Reviewer #1: Partly

Reviewer #2: Partly

Reviewer #3: Yes

2. Has the statistical analysis been performed appropriately and rigorously? 

Reviewer #1: N/A

Reviewer #2: I Don't Know

Reviewer #3: Yes

3. Have the authors made all data underlying the findings in their manuscript fully available?

Reviewer #1: Yes

Reviewer #2: Yes

Reviewer #3: No

4. Is the manuscript presented in an intelligible fashion and written in standard English?

Reviewer #1: No

Reviewer #2: No

Reviewer #3: Yes

5. Review Comments to the Author

Reviewer #1: Dear authors, thank you for the possibility to review this very interesting paper. I find it very relevant for publication as objective parameters of testing always expand knowledge of subjective questionnaires. Yet, there are some remarks I would find really important to add before publication as they would strengthen the scientific standards of your work:

Please report effect sizes, at abstract as well as at result’s section. Please discuss the very important limitation of not using an active control group as literature suggests the effects could also occur through other positive interventions. Also please discuss thoroughly why you exclude women and state this as limitation. I would find it very helpful to have an extra section strengths and limitations at an extra discussion’s section and not mixed at results’ section.

Please make clear why you decide to not call it yoga nidra meditation as a lot of other literature would do so.

There are some small spelling mistakes I have marked at the document attached. There are also some sections where there are words missing to complete a sentence or when a technical term needs to be explained for the reader's better understanding. I wouldn't use that many statistical values at abstract but rather state very clear your results and effects sizes / meaningfulness. Also state limitations as an outlook here.

Please consider APA guidelines for table e.g. table 1 and use dois for your literature section whereever possible.

All the best!

Reviewer #2: Although this manuscript reflects a great amount of work and important findings, there are a number of glaring omissions that significantly reduce my enthusiasm for this study because a reviewer is unable to adequately gauge its methodological rigor.

1. The English used is sub-standard and negatively affects readability. The manuscript needs a thorough review for grammatical issues.

2. The authors assume a familiarity with the yoga nidra technique and some of the methodological details of the study that appears unwarranted and makes it difficult to evaluate. For example, how is yoga nidra different from other yoga and meditative techniques? What did supervised training consist of?

3. No validity data is given for the cognitive and sleep outcome measures. Has research been conducted on the subtests of the CTB? Explain in more detail about the PSG and EEG measures and why they were specifically chosen to evaluate sleep in this study.

Reviewer #3: The paper offers a compelling exploration of the cognitive and sleep-related consequences of Yoga Nidra practice, with the intention of addressing a knowledge gap about the measurable outcomes of this particular practice. The novelty of the study comes in its investigation of a relatively unexplored element of the effects of Yoga Nidra. The manuscript exhibits the author's comprehension of pertinent scholarly works, citing the stress-reducing advantages of Yoga Nidra and its ability to enhance sleep quality. Nevertheless, it is imperative to incorporate citations to contemporary research papers that specifically examine the cognitive impacts of Yoga Nidra.

The study demonstrates a strong methodological framework, incorporating thorough evaluations of cognitive function and sleep patterns using a range of tasks and polysomnography techniques. The selected methodologies are in accordance with the aims of the study, and the statistical analysis provides evidence in favor of the stated outcomes. Nevertheless, it would be beneficial to improve methodological clarity by providing more comprehensive explanations of the cognition testing battery and EEG analysis methods.

The findings are presented in a manner that effectively demonstrates the observed enhancements in cognitive tasks and sleep parameters subsequent to the practice of Yoga Nidra. The conclusions reached are consistent with the findings and discussion of the study. The article demonstrates the significance it holds for future research and the integration of complementary and alternative medicine (CAM) practices into healthcare initiatives. However, further elaboration on the study's wider societal implications and potential limitations would augment the manuscript's comprehensiveness.

The text has a generally well-structured organization, while particular portions might benefit from more refinement to enhance overall readability. The linguistic style employed is inappropriate for a scholarly readership since it effectively presents the data analysis and findings in a manner that is not easily understood. In general, the paper provides significant insights into the impact of Yoga Nidra practice on cognition and sleep, thereby enhancing the comprehension of the potential advantages of complementary and alternative medicine (CAM) techniques. In order to enhance the quality of the paper, it is suggested that a more thorough incorporation of contemporary scholarly works be undertaken, along with certain modifications to improve the clarity of the methodology and the explanation of the consequences. Authors are requested in their discussion to include the following references: https://doi.org/10.3390/bioengineering10020249, https://cdn.techscience.cn/files/jnm/2023/TSP_JNM-5-1/TSP_JNM_37583/TSP_JNM_37583.pdf

6. PLOS authors have the option to publish the peer review history of their article (what does this mean?). If published, this will include your full peer review and any attached files.

Reviewer #1: No

Reviewer #2: No

Reviewer #3: No

---

## [Author Response · Author response to Decision Letter 0]

27 Sep 2023

Dear Editor,

The response to reviewer's comments letter has been uploaded with the re-submission.

Regards

---

## [Decision Letter · Decision Letter 1]

16 Oct 2023

PONE-D-23-21201R1Improved Sleep, Cognitive Processing and Enhanced Learning and Memory Task Accuracy with Yoga Nidra Practice in NovicesPLOS ONE

Dear Dr. Datta,

Thank you for submitting your manuscript to PLOS ONE. After careful consideration, we feel that it has merit but does not fully meet PLOS ONE’s publication criteria as it currently stands. Therefore, we invite you to submit a revised version of the manuscript that addresses the points raised during the review process.

A few additional minor revisions requested. We are getting closer.

We look forward to receiving your revised manuscript.

Kind regards,

Kallol Kumar Bhattacharyya, MBBS MA PhD

Academic Editor

PLOS ONE

Journal Requirements:

Reviewers' comments:

Reviewer's Responses to Questions

**Comments to the Author**

1. If the authors have adequately addressed your comments raised in a previous round of review and you feel that this manuscript is now acceptable for publication, you may indicate that here to bypass the “Comments to the Author” section, enter your conflict of interest statement in the “Confidential to Editor” section, and submit your "Accept" recommendation.

Reviewer #1: All comments have been addressed

Reviewer #3: All comments have been addressed

2. Is the manuscript technically sound, and do the data support the conclusions?

Reviewer #1: Yes

Reviewer #3: Yes

3. Has the statistical analysis been performed appropriately and rigorously? 

Reviewer #1: Yes

Reviewer #3: Yes

4. Have the authors made all data underlying the findings in their manuscript fully available?

Reviewer #1: Yes

Reviewer #3: Yes

5. Is the manuscript presented in an intelligible fashion and written in standard English?

Reviewer #1: Yes

Reviewer #3: No

6. Review Comments to the Author

Reviewer #1: Dear author, thank you for your remarks.

- statistical abbreviations have to be italic (APA)

- lines 57-59, this is not precise, one part of yn is pratyahara but not all of it, and there are studies that highlight alpha waves !

- line 68/69, there is the same source two times, online first and print versions are the same. please doublecheck your literature if there are more doubles like this

- line 70, there are two dots. Also at 273

- 84 ff important information has always to appear also at running text, e.g. how many males took part in this study; 89 ff how long was the yn

- did you perform a power analysis to find out about the sufficient sample size for your study? please report about the power of your study, at least at discussion's section about strengths/weaknesses - there ist still missing a comment on the value of active control groups!

- line 180 (also 248) numbers are written as numbers when above 10 (APA); how many is "some"?

- why do you report different effect sizes? d, r and eta - please either choose one or explain why you choose three and differentiate the thresholds of small, medium, big either way

Thank you and kind regards

Reviewer #3: The article is well written. all technical comments were included in the revised version. A proper English review is requested before publication.

7. PLOS authors have the option to publish the peer review history of their article (what does this mean?). If published, this will include your full peer review and any attached files.

Reviewer #1: No

Reviewer #3: No

---

## [Author Response · Author response to Decision Letter 1]

21 Oct 2023

Dear Reviewers,

Thanks for your comments and the journal office email dated 16 Oct regarding revision in the submitted manuscript.

The response to reviewers dated 21 Oct 2023 has been uploaded and necessary modifications done in the revised manuscript.

Regards

Karuna Datta

---

## [Decision Letter · Decision Letter 2]

7 Nov 2023

Improved Sleep, Cognitive Processing and Enhanced Learning and Memory Task Accuracy with Yoga Nidra Practice in Novices

PONE-D-23-21201R2

Dear Dr. Datta,

We’re pleased to inform you that your manuscript has been judged scientifically suitable for publication and will be formally accepted for publication once it meets all outstanding technical requirements.

Kind regards,

Kallol Kumar Bhattacharyya, MBBS MA PhD

Academic Editor

PLOS ONE

Additional Editor Comments (optional): As the reviewer suggested, please write consistently yoga nidra, not yoga-nidra during proofreading or when submitting your final version of the manuscript.

Reviewers' comments:

Reviewer's Responses to Questions

**Comments to the Author**

1. If the authors have adequately addressed your comments raised in a previous round of review and you feel that this manuscript is now acceptable for publication, you may indicate that here to bypass the “Comments to the Author” section, enter your conflict of interest statement in the “Confidential to Editor” section, and submit your "Accept" recommendation.

Reviewer #1: All comments have been addressed

2. Is the manuscript technically sound, and do the data support the conclusions?

Reviewer #1: Yes

3. Has the statistical analysis been performed appropriately and rigorously? 

Reviewer #1: Yes

4. Have the authors made all data underlying the findings in their manuscript fully available?

Reviewer #1: Yes

5. Is the manuscript presented in an intelligible fashion and written in standard English?

Reviewer #1: Yes

6. Review Comments to the Author

Reviewer #1: - Please write consistently yoga nidra, not yoga-nidra

- Please double check your literature and add “dois” whereever available

7. PLOS authors have the option to publish the peer review history of their article (what does this mean?). If published, this will include your full peer review and any attached files.

Reviewer #1: No

---

## [Editor Report · Acceptance letter]

20 Nov 2023

PONE-D-23-21201R2 

Improved Sleep, Cognitive Processing and Enhanced Learning and Memory Task Accuracy with Yoga Nidra Practice in Novices 

Dear Dr. Datta:

I'm pleased to inform you that your manuscript has been deemed suitable for publication in PLOS ONE. Congratulations! Your manuscript is now with our production department. 

Kind regards, 

on behalf of

Dr. Kallol Kumar Bhattacharyya 

Academic Editor

PLOS ONE